# The green, gold grass of home: Introducing open access in universities in Norway

**Lars Wenaas** *, **Magnus Gulbrandsen**

TIK Centre for Technology, Innovation and Culture, University of Oslo, Oslo, Norway

☯ These authors contributed equally to this work.
* lars.wenaas@sikt.no

**Data Availability Statement:** The datafile is available from the DataverseNO database (https://doi.org/10.18710/IQGCHB).

## Abstract

In this paper, we investigate how open access is reflected and implemented in all Norwegian universities and how they responded to national policy developments for open access in the period 2009–2021. We analyse how the universities adapted arguments for the three core missions of the universities–research, education, and societal impact–when they reacted to increasing pressures to facilitate open access. Our analysis is based on 182 institutional strategy documents, open access policies and annual reports. When considering the profile of the institutional policies and the explicit referrals to national policies, we find there is a great deal of homogeneity between Norwegian universities, and they are mostly aligned with national policy. Open access is connected to the third mission in all university strategies, but often in a very general manner and without documenting benefits for non-academic users. We find limited emphasis on open access as advantageous for education. All universities show commitment to open access, and several can be described as proactive as they tie it to different types of local incentives. Development over time suggests more mature and institutionalised polices that do not challenge what we may call the academic heartland and its core value of academic freedom, including where and how to publish. We propose a framework for analysing similar institutionalisation processes with three main dimensions: mimesis, adaptation/integration with existing practices, and maturation/commitment.

## Introduction

How do universities implement and institutionalise new activities when these activities are rooted in external expectations and demands? An important recent expectation concerns open access publishing where governments and research councils have formulated increasingly strict policies and introduced demands in university budget documents and research council contracts. In Norway, the Ministry of Research and Education launched national guidelines for open access in 2017 with the goal of full coverage within 2024. The push towards open access is most clearly seen in *Plan S*, a policy launched in 2018 by a federation of European research councils [1]. How universities implement changes as response to external policy, is considered crucial for understanding the move to more openness [2, 3].

Earlier studies of institutionalisation processes in higher education institutions have in particular concentrated on the so-called 'third mission' [cf. 4], which encompasses

**Funding:** LW is financed by the Research Council Norway (RCN-project: 272456) and the 'Oslo Institute for Research on the Impact of Science' (OSIRIS, RCN-project:256240/O30) MG is financed by the Research Council Norway ('Oslo Institute for Research on the Impact of Science', OSIRIS, RCN-project:256240/O30) The funder had no role in study design, data collection and analysis, decision to publish, or preparation of the manuscript.

**Competing interests:** The authors of this manuscript have the following competing interests: Lars Wenaas is a PhD-candidate at the TIK-center at the University of Oslo while holding a position at 'Sikt', a governmental body reporting to the ministry of education and research in Norway. The position is a part of the department with responsibility of coordinating open access affairs in Norway. This does not alter our adherence to PLOS ONE policies on sharing data and materials.

commercialisation and societal engagement [e.g. 5–8]. Important findings in this literature include the characteristics of the complex and long-term process whereby new activities become taken for granted through changes in internal activities [5]. Investigations of institutionalisation of the third mission mostly start from an assumption that this represents a tension or a mismatch with education and research activities (the first and second missions).

The introduction of open access is most likely different from many universities' experiences of the third mission. Openness is tied to a key practice in universities–publishing research articles–and it is principally something that can support all three institutional missions by making research more widely available to students, colleagues, and society at large. Policy rationales for open access also refer to the third mission and societal usefulness [e.g. 9, 10]. Our analysis is therefore of a process that does not represent a clear clash of institutional logics or problematic externally pushed practices.

However, introducing open access may touch upon sensitive issues like academic freedom and intellectual ownership, and resistance to open access polices indicate that they are not uncontroversial. Even though most researchers have a positive attitude towards the principle of open access, policy details may cause harsh debates. Plan S is a good example, where research funded by the signatories demands immediate free access. Gold open access–publications in non-subscription-based journals is prioritised here, not 'green open access' which is tied to depositing scientific texts from subscription-based journals in repositories, with dissemination mostly after a publisher-imposed embargo. Critics have attacked its approach [11, 12] and issued worries over interference with academic freedom and limitations in individual choice of journals [13].

We nevertheless see these concerns as minor compared to extensive emphasis on somewhat controversial third mission tasks like commercialisation for overburdened institutions [e.g. 14, 15]. Open access may require *maturation* and various changes in standards and practices [cf. 5]. Our paper seeks to add to the literature on institutionalisation processes and its emphasis on understanding and conceptualising institutional changes in universities, but in a context where such change may have strong internal champions. Our empirical investigation of institutionalising open access is a theme that, as far as we have found, is not covered in earlier studies.

Based on these considerations, we ask the following exploratory research question: *how did Norwegian universities respond to expectations about open access, and how were these implemented in institutional-internal policies?* We see university and national level documents as an appropriate data source for answering this question [as in 5, 6, 8]. We are particularly interested in how open access is tied to existing university missions, which has proved a useful heuristic in earlier studies of institutional change in universities. Our expectations are that open access is most easily tied to the mission of research and its norms of openness and integrity, and that there may be variance between institutions due to their heterogeneity. Norway is a very interesting case as an open access forerunner, and our analysis includes all the country's universities.

The paper is structured as follows. First, we review relevant literature with emphasis on open access and its relation to the three missions, and studies of institutionalisation of policies in universities. We then proceed with an analysis of Norwegian universities' response to the policies based on their official documents, exploring developments, similarities, and differences between the ten Norwegian universities. Our final section contains practical implications and theoretical considerations about institutionalisation processes.

## Review of relevant literature and the Norwegian national policy development

The literature on open access varies considerably and covers topics like publishing business models and costs [16–18], policies and prevalence studies [2, 19–21], researchers' attitudes

towards open access [22–24], predatory journals [25, 26] and different impacts of open access [27, 28], particularly academic impact [see 29]. These investigations are primarily empirical, and we have not been able to identify any studies on open access that use institutional theory, the study will therefore draw on the institutionalisation of the third mission. This is relevant as a parallel case in terms of institutionalisation and directly linked to open access in science policy. This relationship is also a topic in the last section of the literature review, where we give a brief account of the development of national open access policies in Norway.

## Open access and its conceptual ties to university missions

Open access emerged not only as a response to access problems, but as a response to severe economic challenges for universities. The 'serial crisis' refers to the situation in the late 1980s when subscription prices grew while library budgets declined, which created an unsustainable economic situation for higher education institutions and their libraries, forcing them to reduce journal subscriptions [30, 31]. Digitalisation and internet-related opportunities made publishers' marginal cost drop significantly, and there was hope that this would provide a solution to the serial crisis. However, given the nature of academic journals as a non-substitutable good, restricted access remained the norm.

The serial crisis paved way for the formalisation of open access through a series of declarations known as the Berlin-Bethesda-Budapest initiatives in 2002–2003, of which the Budapest declaration is the most widely cited. The Budapest declaration stated the goal of access to the literature by 'all scientists, scholars, teachers, students' which will 'accelerate research, enrich education', while other user groups where labelled as 'other curious minds' rather than industry or other specific non-academic actors [32].

Open access comes in three main flavours. *Gold* open access is tied to journals where licences enable free and extensive user rights. *Green* open access encompasses subscription-based journals where articles are put in a repository with subsequent open dissemination after an embargo of usually 6–12 months. Finally, *hybrid* refers to single articles in subscription-based journals bought free through article processing charges (APC) for immediate release. These are licenced in an identical or similar fashion to articles in gold open access journals. While the first open access declarations saw green and gold as complementary viable options, hybrid entered the scene later [33]. Furthermore, hybrid has been discouraged for economic reasons. The biggest challenge is double-dipping, where publishers charge both processing charges and subscription fees. Hybrid is therefore ineligible in most financing schemes unless part of agreements designed to push publishers into converting their journals to gold open access.

Open access reflects the ideal-normative approach of Merton setting up five central norms of science: communalism, universalism, disinterestedness, originality and scepticism ('organised scepticism' in the first version) [34]. Open access has been viewed as a direct translation of communalism [35], expressing how researchers share discoveries and give up intellectual property in exchange for social recognition [34]. Universalism also denotes the imperative of sharing research results widely. Studies argue for both a download advantage and citation advantage from open access [27], although the latter is connected to the type of open access that the author selects. Generally, green and hybrid open access show larger citation advantages than closed access, while gold open access advantages are less clear and can be disputed [19].

Academic impact is connected to the journals' role in the reward system of science, where the most prestigious and highly ranked journals are often established and prestigious 'legacy' journals, as opposed to comparatively new gold open access journals. Journal rankings and

their role in the academic reward system are often considered main barriers for the uptake of gold open access publishing [36]. The uptake of open access varies between the disciplines both in terms of levels and types. The medical sciences have the highest levels of open access, closely followed by the natural sciences, while the social sciences are lower but remain above the levels for the humanities [20]. The different modalities of open access also vary due to different structural and institutional factors, polices and other sociotechnical aspects (ibid.)

Although open access is tied to core research norms, it involves benefits for education–the first mission–since academic texts are key elements in curricula. Open educational resources include not only open access but the wider idea of open scholarship, access to educational opportunities and the adoption of open learning strategies [37]. A UK study found that 38% of journal articles licensed centrally by a collective management organisation and used for teaching purposes, were (some form of) open access, yet still a part of the collective licensing arrangement [38]. Although open access would benefit universities, especially with respect to the articles not licensed centrally, the largest potential is to give students open access to monographs and textbooks [38]. There is also a great potential in supporting sustainable lifelong learning for user groups like medical patients and their networks, health advocates and NGOs [27].

Considering benefits for the third mission, the scientific literature is a particular codified output from research work, which potentially is useful for both private and public sector actors. For example, it has been argued that lack of access to the scientific literature for small and medium-sized biotech firms hampers innovation [39], and open access is also desired by researchers in the private sector [40]. However, empirical support for benefits is rather limited. A review of the economic benefits of open science found indications of cost savings with open access, but evidence for economic returns are still considered patchy and diverse [41]. A few studies indicate that openly available articles have an advantage compared to the paywalled literature when considering patents. The U.S. National Institutes of Health's 2008 open access policy was associated with an increase of 12–27% in patents citing its funded research [42]. Large pharmaceutical companies more often cite open access literature and publish more in open access journals [43]. Country reports from the UK, Netherlands and Denmark conclude that private firms save costs, but otherwise present little evidence for innovation benefits [44–46].

A more principal argument sees science as a public good and scientific knowledge as something that should be free and under little restriction [47]. Public good is a term borrowed from economics [48] and is also commonly applied in the context of open science [49–52]. For example, Dalrymple maintains that 'Scientific knowledge in its pure form is a classic public good. It is a keystone for innovation, and in its more applied forms is a basic component of our economy' [47, p. 35]. Although the question of whether open access will increase private sector innovation is both interesting and, in our view lacks empirical investigation, we proceed with the observation that policies have adopted the argument that access to the scientific literature is important for different forms of innovation among private and public actors.

Policies and funder mandates are considered the main drivers of open access the last decade [2], where the devil hidden in policy details influences their effect [21]. When, how, and how much of the literature is made open is connected to opt-out mechanisms, mandatory depositing and whether incentives and research evaluations contain open access points [53].

While the idea that the academic literature should be freely available is broadly shared, there is substantial disagreement on the best approach to achieve open access [2]. Policy preference may conflict with autonomy and other key academic values; the push for immediate open access by strong mandates, as in the case of Plan S, may exclude subscription based

journals from the researchers' options [54]. This is therefore seen as an infringement on academic freedom [13].

Studies of open access polices have mostly been concerned with what constitutes an effective policy and their performance [e.g. 21], rather than institutional responses and practices. A study that moves beyond quantitative effect-analysis is a Dutch report that found a top-down induced growth in open access driven by national policies [55]. An Australian investigation, noting that there is no unified national policy, found that half of the country's 40 universities had no open access policy, and those that had, exhibited a wide variety in their approach in terms of definitions, depositing time and sanctions [56]. Institutional polices were introduced over a period of 17 years with some references to generic benefits to society. Furthermore, references to the two major Australian funders were found in all 20 policies, with a high degree of compliance (ibid).

## Institutionalisation of new missions and practices in universities

We have seen that the adoption of open access in science policy is intimately connected to beliefs about benefits to the wider society. Policy documents argue that open access will lead to better utilisation of research-based knowledge in society, both in Norway [57] and internationally [9, 58]. The EU policies in particular frame open access and open data as fuel for the innovation engine, ascribing a commercial third mission logic to open access. The third mission is therefore not only useful as an example of the possible institutionalisation of new tasks in universities–it has become an important rationale for open access which connects to an increased interest in university-industry relations and how universities may contribute to economic and social development.

This interest has produced investigations of how universities introduce and implement new practices. A frequent framing is Clark's [59] investigation of 'entrepreneurial universities' that have transformed themselves and developed significant support of the third mission [cf. 4] of innovation and societal engagement. Noticing an increasing gap between demands to universities and their capacity for response, Clark found that there are several aspects to successful institutional transformation. This includes the universities' ability to *integrate* new activities with existing ones, and their use of new activities to protect *the academic heartland* of collegiality, autonomy, academic freedom and achievement. The academic heartland thus conceptualises traditional academic values, which normally are sought protected when confronted with new demands, which can make changes more likely to fail.

Although Clark's and other's empirical investigations have shown that many universities manage to institutionalise new practices without threatening the heartland, severe challenges have been identified. Some universities, even when they are heavily engaged in various forms of societal engagement, have struggled to meet policymakers' expectations of third mission activities [7, 8, 14]. Responses to external expectations have often failed to become integrated with existing practices and strengths, and therefore remained peripheral or in tension with education and research missions. Often new external demands are poorly suited to higher education institutions' unique profile [6, 14]. The blame is often put on external policies rather than institutional agency, for example that policies push some aspects of the third mission, commercialisation in particular, in favour of others that may engage all disciplines and staff members [7, 8, 14]. Another challenge is that uniform demands ('one size fits all' policies) to a heterogeneous sector and competitive funding based on problematic indicators create pressures that universities struggle to integrate effectively [6]. This leads to unique responses because each 'university is the product of a distinct process of social, economic and intellectual development' (ibid. p. 738). Universities are thus subject to isomorphic forces [cf. 60] or heterogeneous institutional logics [e.g. 61] that need to be resolved in one way or another [6].

In an in-depth historical study of Stanford University, Colyvas & Powell [5] show how an interplay between top-down and bottom-up internal initiatives contributed to the institutionalisation of technology transfer practices over time. They use this empirical analysis to set up a conceptual framework for understanding three interrelated dimensions of such processes. First, *institutionalisation* concerns (changes in) organisational structure, practical action plans and ensuring reproduction of the new activity through learning, training programmes and dissemination. Second, *legitimacy* refers to how new standards develop, how norms of what is appropriate behaviour are enforced and how boundaries to existing practices are remade. Third, *taken-for-grantedness* implies gradual changes in practices and roles and the emergence of new categories. Colyvas & Powell furthermore indicate three states for each of the dimensions: low, medium and high. A low state can be related to symbols and vocabularies drawn from external sources, medium is represented by institutional-specific ways of talking about the issue in question, and high is when a rich, local language becomes widely accepted.

We initially argued that the 'openness' of open access may be more in line with research norms. However, tensions can still emerge at the intersection of research, the third mission and open access, not least due to how the outside pressure in demands is put on universities.

## The development of national open access policy in Norway

A core interest in this study is how universities respond to government expectations, which comes in the form of policies on open access. One of the first official statements on open access in Norway appeared in a 2005 White Paper, framing it as a response to the increasingly high subscription fees for academic libraries, which could hamper access for researchers, particularly in developing countries. The ministry applauded the introduction of open access journals and repositories for green open access [62]. In 2008, the Ministry asked the interest organisation of universities and colleges for advice on how to implement open access, which led to the establishment of a collaborative infrastructure project [63]. Future policies were therefore based on the advice of higher education institutions with important contributions from the universities to the initial national open access policy.

A 2009 White Paper on science defined open access as one of nine central policy topics and argued for the importance of this for researchers, firms and the general public [64]. Even though open access was seen as advantageous, it was not unequivocally pushed: 'in principle, the government believes that all public research should be openly available, unless other considerations prevent it' [64. p. 125] and 'researchers should, however, not be required to deposit articles in open repositories if publishers do not allow this.' [64, p. 127]. These reservations challenged the general point made that open access largely could be realised without compromising academic freedom and copyright [64]. The infrastructure project was made permanent and assigned to a newly established organisation Cristin (Current Research Information System in Norway). The Research Council Norway (RCN) issued a policy the same year arguing for open access so that 'the private sector . . . easily can access published scientific results' [65]. This policy and the 2009 White Paper served to establish the idea of open access; they also highlighted green open access and saw depositing in repositories with embargo as sufficient.

Following a regular four-year cycle for White Papers on science policy, the 2013 White Paper reaffirmed the principles of open access, largely with the same arguments [66]. Again, academic freedom was given priority to considerations of access. The policy also condoned 'the taxpayer argument'–an argument that it is unreasonable for publicly financed research output not to be openly available, as the taxpayer has already financed it [66]. The same white paper gave Cristin the responsibility for developing indicators for open access similar to other

science policy indicators. In 2014, RCN revised its open access policy and included funds for gold open access publishing.

The Ministry of Education and Research launched its national guidelines for open access in 2017 [67]. These were based on a report from an expert group of researchers [68] with a goal of full open access by 2024. The guidelines affirmed Norway's continued role as a 'cautious forerunner' in the pursuit of open access for the benefit of research, the wider society and innovation [68]. They prescribed gold open access as the main ambition but allowed for green open access without specifying a maximum embargo. This signalled a shift in preference for immediate open access, including hybrid when part of a transformative arrangement. Depositing was made mandatory for all articles, and the ministry maintained that when the system 'allowed for it', only deposited articles would be considered eligible for the results-based funding system, thus connecting open access to future financial incentives.

Norway's only research council RCN was one of the original signatories of Plan S [1], a policy with a non-negotiable preference for immediate open access. The first version allowed only gold immediate open access publications, a requirement which was softened in later revisions allowing immediate green open access yet remained stricter than the national guidelines. Although Plan S primarily concerned research council funded projects, the policy was launched with a significant political investment from the Norwegian Ministry. A 2018 White Paper signalled a tempo shift and portrayed Norway as a forerunner in the pursuit of open access along with the Netherlands, the UK and Germany, and further aligned Norwegian policy with European polices on open science [69]. Other initiatives have followed, for example a 2019 report on creative commons licences in academic publishing discussing whether demands on the researchers to choose creative commons is at odds with academic freedom [70].

With the Norwegian policy development as background, our analysis will use official university documents, similar to Kitagawa et al. [6], Colyvas & Powell [5] and Wakeling et al. [56]. The main concepts we bring into the analysis concerns how institutional polices and strategies handle open access and its relationship with the three missions and the 'academic heartland' and further how this reflects or contrasts with national policy. We explore whether the implementation of open access is a process that can be separated into institutionalisation sub-dimensions that develop into distinct states or stages over time.

## Context, methodology and data

Institutional open access efforts include setting up repositories, initiating dedicated open access publishing funds, strengthening libraries' efforts in rights clearance, or hosting open access journals. These efforts seem to have proven successful when viewing the increase in open access publications at all 10 Norwegian universities. Open access publications have shown a large growth over the last decade and now represents the majority. A plot of the growth in open access at Norwegian universities can be found in S1 Fig. Norway has ten universities per 2022, and we have included all of them in our investigation. Starting from low numbers, they have increased their total share of open access (not including deposited documents) to 71.1% on average in 2020 [71]. Notable patterns are jumps in green open access for institutions who mandated depositing following the launch of National guidelines in 2017, and a similar increase in hybrid after the introduction of Plan S-influenced publish-and-read deals in 2018. Gold open access has further grown steadily in all institutions throughout the period, backed up by institutional funds. All 10 universities have dedicated funds for supporting gold open access publishing (although the University of Oslo abandoned this in 2019), which in turn are supported by RCN's reimbursement program covering up to 50% of gold

open access article processing charges. The government and Norwegian universities have thus collectively maintained a financial infrastructure in support of gold open access.

While the shares and types of open access publishing, funds, transformative agreements, and other infrastructural factors certainly are interesting aspects, we are primarily interested in understanding how this notable change has happened within the Norwegian universities. Our starting point shares the perspective of Montgomery et.al. [3]: policies and strategies of universities act as signals of intent and are proxies for organisational and institutional support of change, which means that official university documents can be a good indication of the institutionalisation of open access. These documents are subject to formal and mostly transparent decision-making processes and communicate to both staff members and external stakeholders. For our research question–investigating institutional responses to open access policies over time–official documents likely hold high validity. Such documents can be a valuable data source in qualitative, inductive analysis of the unique reality of specific actors [72]. Grounded theory approaches argue that documents are comparable to interviews and other fieldwork [73], and they can be coded and analysed in a stepwise and rigorous way [74].

To capture variability in institutional responses, we have primarily investigated three types of documents: institutional open access polices, main strategy documents (institutional priorities over a multiyear period) and annual reports from all ten Norwegian universities, in total 182 documents. The timeframe is set to the period of 2009–2021 to include the starting point of the first national policy. Most documents have been downloaded from institutional web sites, but to capture changes and development over time, the list has been complemented with previously issued strategy documents collected from web archives. OsloMet and the University of South-Eastern Norway were formally accredited as universities in 2018, we have therefore used documents from the preceding institutions (colleges) as proxies to provide a more comprehensive timeline.

We have included institutional open access resource webpages, minutes of university board meetings where possible and ROARMAP, a database containing open access policies worldwide, including several Norwegian, as these often elaborate on policies in more detail. We have also included consultation letters from the universities that were a part of the formal hearing process for the 'National guidelines for open access to research articles' in 2016.

Annual reports are comprehensive documents, many around 150 pages, highlighting the breadth of activities and different sources of pride for the institutions. For many aspects such as research, educational initiatives, regional importance and international partnerships, the documents contain a host of examples with pictures of and interviews with students, staff, stakeholders and project participants. We expect annual reports to include open access indicators when requested in allocation letters by the Ministry of Education and Research. Since 2014, annual reports have been a formal part of the Ministry's steering of the sector, reports are therefore included from 2014 onwards. Table 1 gives an overview of all Norwegian universities and the number of collected documents for each institution.

Our data set and underlying analysis is publicly available: A full list of documents, including links, text excerpts, categorisation and coding can be retrieved at https://doi.org/10.18710/IQGCHB [75].

The analysis started out by each author analysing all documents from two of the universities, writing a report about themes, contexts, relationships, language use and other issues, similar to open coding of qualitative data [74, 76]. There was only minor variation in interpretations, which we take as high intercoder reliability, a term that for us also implied making both authors deeply familiar with the data and improving reflexivity [77]. After agreeing on a final coding scheme, which also had elements of axial coding whereby the initial

**Table 1. Overview of institutions, abbreviations and number of documents.**

| Institution | Abr. | No. of documents |
|---|---|---|
| Norwegian University of Life Sciences | NMBU | 20 |
| University of Bergen | UiB | 21 |
| Nord University | Nord | 18 |
| University of South-Eastern Norway | USN | 13 |
| University of Tromsø | UiT | 16 |
| Oslo Metropolitan University | Oslomet | 21 |
| University of Oslo | UiO | 17 |
| The Norwegian University of Science and Technology | NTNU | 19 |
| University of Stavanger | UiS | 15 |
| University of Agder | UiA | 22 |
| **Total** | | **182** |

categories related to open access and university characteristics were related to one another [76], the rest of the documents were analysed by the first author.

Our open coding stage meant reading the documents carefully to look for passages that explicated university ambitions for open access, which resulted in primary codes related to the following themes:

- The *context* of open access in strategy and policy documents and to what extent open access is seen in conjunction with the three core missions.

- Evidence provided for *benefits* of open access to any of the missions, including codes about the missions themselves.

- The *profile* of local open access policies including open access preference, mandates for depositing and opt-out possibilities.

- *Links and quotes* referring to national policy-documents.

- *Local incentives* ('carrots and sticks') connected to open access.

- The contents and level of *formal reporting* on open access publishing in annual reports.

Axial coding and later analysis were conducted in two main steps:

1. Identifying changes in the role of open access within each university, by comparing revisions of and development in polices and strategies.

2. Investigating the institution's current policy on open access and its alignment with the national policy, particularly the national guidelines in 2017, and to what extent the institution has adapted to, or deviates from national policy.

For the first step of the analysis, we also reconstructed a timeline of the universities' open access polices to look for signs of institutionalisation. While the historical account of open access on the national level is well documented, the institutional level can be more challenging. When strategies and policies are amended, modified, or replaced by revisions, old documents are sometimes removed from the institutional website. Our data is complete with respect to the current situation, but there may be gaps in the historical development even with our access of web archives. We nevertheless claim we have identified the most important milestones in institutional development.

Nine of ten universities have revised their main strategy in the period of analysis. We also found that eight universities have revised their open access policy in the period, six have revisions after the launch of national guidelines in 2017 and one based its 2016 policy on the commissioned report that the guidelines were grounded on.

A strength of our analysis is that it covers all the Norwegian universities, a long time and a comprehensive set of public documents that have been created for other purposes than our investigation. As such they may be seen as more 'objective' or 'unobtrusive' [cf. 72]. A challenge with documents is that they may be biased or inaccurate in various ways (ibid.). Some of the university strategy documents paint a somewhat glossy picture of the institution's ideals, practices, and successes. On the other hand, they also express priorities, and the annual reports to the Ministry are likely highly accurate as their numbers (like publication performance) are easily verified by third parties, and their categories are defined externally.

We use 'depositing' for the act of archiving an article in a repository, but this does not include making the article openly available. 'Opt-out' possibilities are mechanisms that provide the researchers with the option of not depositing an article, or in the case of mandatory depositing, provides the option of not making it open access. 'PAR-deals' refer to consortia Publish-and-Read-deals with publishers, which include the possibility of publishing hybrid open access for authors at member institutions. 'Preference for gold' means that the institution prefers (but does not mandate) publishing in gold open access journals over publishing in subscription-based journals.

## Findings: Institutional developments

The Norwegian universities developed their institution-specific open access policies in different years. We see this timing discrepancy as an indication that open access, even if an increasingly important part of national science policy, was not perceived as an urgent problem that needed an immediate solution. Open access drifted into institutional strategy documents when they were revised.

### The role of open access and the three missions

All ten universities tie openness to research in strategy documents, and they furthermore connect openness to the third mission and societal engagement. However, as the term openness implies, statements are often general, and the strategies do not necessarily apply the term 'open access'. An example is the Norwegian University of Life Sciences (NMBU): 'NMBU will contribute to a knowledge-based and open public debate.', 'have an open and respectful collaboration with the world around us, both with the public and private sector. [..] ensure that funding is used efficiently and purposefully and for the benefit of the whole of society', concluding that 'knowledge must be shared.' These statements support the general idea of open access, and we find that these passages from the main strategy are quoted and operationalised in NMBU's open access policy. The Norwegian University of Science and Technology (NTNU), which did not mention open access in its 2011 strategy, stated that '[the university] disseminates research results openly' in the third mission section of its 2018 strategy. Open access is commonly integrated in the context of third mission activities, outreach and science communication also in other universities' main strategies. The University of Bergen (UiB) underlined societal contact and the role of science in society in their 2011 strategy without invoking open access. In 2016 and 2019, similar strategy statements brought up open access as means of dissemination.

Seven institutions connect open access to the third mission in the sense of innovation and benefits for the private sector. However, innovation is a rather minor part of main strategies,

and they often contain general claims that the private sector will benefit from open access. An example is the University of Oslo (UiO), which emphasised open access and innovation in its 2011 strategy. This was followed by a separate innovation strategy for 2013–2015 that included open access, but the claim was downplayed in the next (current) strategy.

However, we also see examples of strengthened links between open access and innovation. The University of Tromsø (UiT) made generic statements about open access and societal contact in its 2011 strategy, while the most recent strategy portrays UiT as 'a driving force for increased innovation to contribute to business and development in the public sector' due to a 'strong culture for dissemination through open channels of publishing'. NTNU's publishing policy (2014–2020) states, 'we know that scientific publications are a central source for innovation in the private and public sectors'. UiB's open science policy (2020) puts forth a similar argument, referring to technological changes and digitalisation and makes a connection between openness in science and open innovation.

Open access is thereby more frequently integrated into statements about the universities' third mission than statements about research activities (second mission). While there is an innate connection between open access and research, the context of research in strategies is often excellence or international collaboration. Open access is instead found in lower-level strategy documents and internal web pages about research.

None of the main strategies mention open access as supportive of education, while only two general open science policies cover educational resources. NTNU's open science policy 'includes all results from research, education and dissemination activities' and UiB's open science policy includes a section on open educational resources to 'facilitate employees' sharing of educational resources and make educational resources openly available in cases where it does not conflict with professional or legal considerations.' Eight institutions do include students in their open access policy, primarily through mandating or encouraging students to deposit master theses in the institutional repository.

We also find that although the universities highlight the importance of open access for the third mission and various societal aims (democracy, critical debate, innovation etc.), we find no examples or documentation in strategies or annual reports of the benefits of open access to a wider audience that could lend legitimacy to the policies. Open access accounts (numbers, graphs) in annual reports are presented unsystematically and summarily, if at all. Annual reports are, however, used for other open access purposes, for instance introducing goals and expectations for the next period, (plans for) revisions of policies, or simply stating the importance of open access. We also note an interesting rhetoric twist in NTNU's narrative in the annual report for 2020, which reports the decreasing shares of closed articles instead of the overall increasing open access numbers.

Table 2 shows the presence of any reporting of numbers on growth and prevalence of open access in annual reports, either in the form of tables, graphs or writing. We have not included reporting on other aspects related to open access, for example financial aspects (costs of open access) or information campaigns.

Eight institutions have revised their open access policies since 2009, primarily stating a preference of gold open access, even if the policies default to green open access and have opt-outs for green. Furthermore, all universities require their employees to deposit research articles in the institutional repository, with three institutions supplying opt-out-possibilities.

There are interesting differences in compliance measures. UiO and the University of Stavanger (UiS) have chosen a formal approach by incorporating mandatory depositing in employment contracts following considerations tied to academic freedom and intellectual property rights (IPR). The other universities have not put this aspect into employment contracts.

**Table 2. Timeline of annual reports.** Marked cells signify whether the report incorporates any numbers on the growth and prevalence of open access.

| | 2014 | 2015 | 2016 | 2017 | 2018 | 2019 | 2020 |
|---|---|---|---|---|---|---|---|
| **NMBU** | | | | | | | |
| **NTNU** | | | | X | X | X | X |
| **OsloMet** | | | | | | X | X |
| **UiA** | X | | | | | X | |
| **UiB** | X | | | | | | X |
| **Nord** | X | | X | | | | |
| **UiO** | | | | | X | X | X |
| **UiS** | | | | | | | |
| **UiT** | X | | | | | X | |
| **USN** | | | | X | X | X | X |

Over time, we find that open access policies use formulations we interpret as stronger with more insisting language. For example, the NTNU policy of 2014 stated that 'knowledge ought to be easily available', while the 2021 revision replaced 'ought to be' with 'must'. Similarly, the former UiB policy 'asked' all employees to deposit a version of scientific articles, while the current policy is more insistent.

In some cases, there is a shift in the responsibility to fulfil open access obligations. NMBU's 2014 policy stated that 'knowledge about open access is to be develop through information and training aimed at university employees', while the 2019 version declared that the researchers are responsible for acquaintance and compliance with open access regulations.

Four institutions have implemented incentives. UiT and USN have incentives directed at the individual researcher since approval of sabbaticals, tenure and promotion will take open access into consideration. UiO and UiB have anticipated the forthcoming arrangement in the national guidelines and as such place negative incentives at the department level. Scientific articles are not eligible for the national reimbursement program unless they are deposited, regardless of open access type. While this primarily has economic consequences at the institutional level, trickle-down arrangements that allocates funds back to the departments incentivise local action [78]. UiO has further announced that open science is one of the elements which should 'change the routines for evaluation of promotion and tenure'.

OsloMet, one of the forerunners on open access in Norway, introduced incentives for depositing as early as in 2010, but later removed them as a part of the institution's merger process in 2018. We do not know whether this is due to legal aspects in the merger process, or whether incentives no longer were deemed necessary and depositing behaviour was seen as institutionalised. The latter may be a reasonable assumption, since OsloMet had over 95% of all articles deposited in its repository as early as 2010 [79].

Six institutions discussed open access and the boundaries to academic freedom and intellectual property rights (IPR) in their documents. All of them prioritised academic freedom and legal considerations with respect to ownership of texts and emphasised the individual's journal selection rights. This is mostly statements about granted rights; Nord University typically confirmed that 'as a researcher, you have the right to choose publishing outlet when publishing your work'.

We summarise the timeline of the broader integration of open access at the universities in Fig 1, which illustrates if and how the institutional strategies integrate open access generally and/or with either of the three missions. We use colour codes for three levels of open access engagement (low, medium, high) in strategies, the same colour codes are used to signal whether policies have evolved in terms of language use and the use of mandates. In addition, we indicate whether polices prefer immediate gold open access.

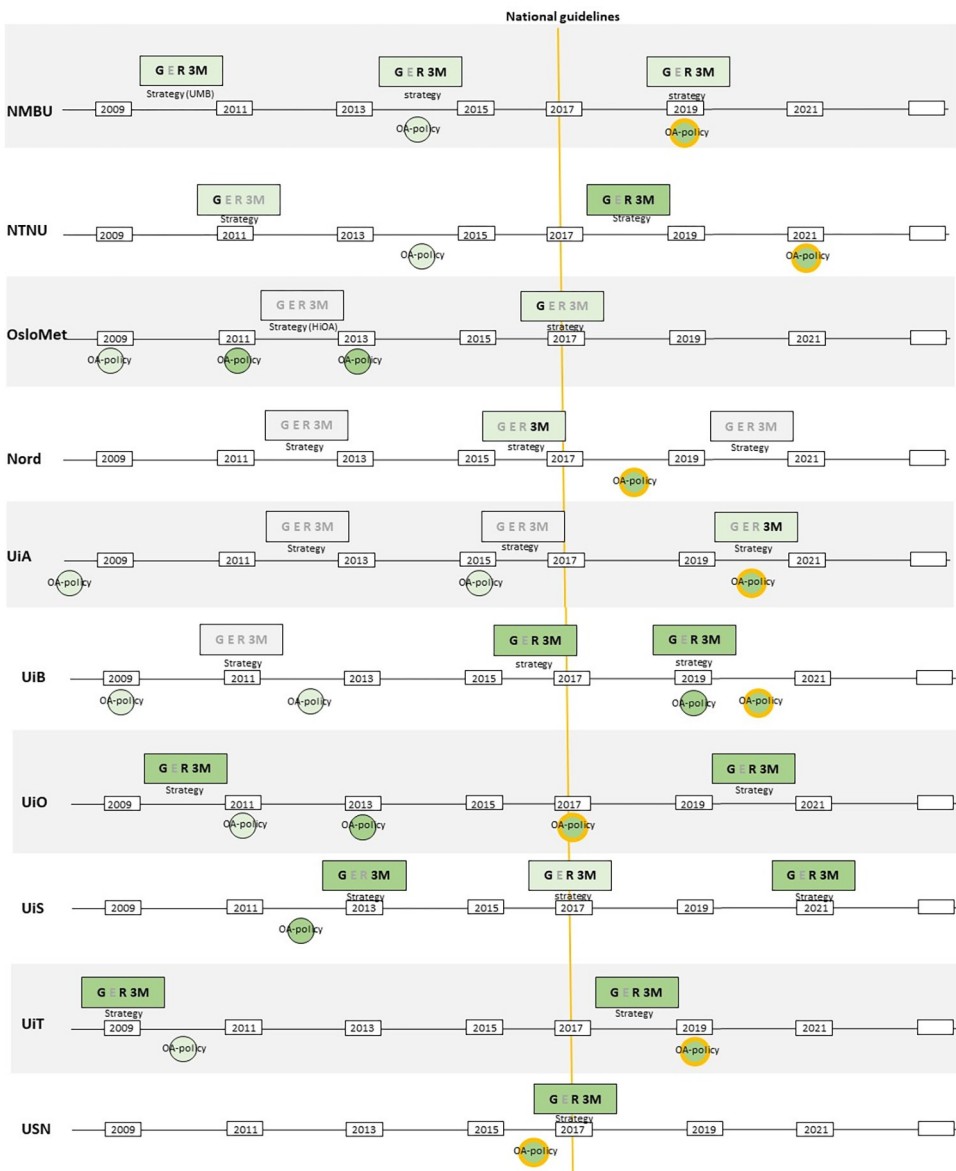

**Fig 1. Summary of the development of the ten Norwegian universities' strategies and policies.** Summary of the development of the Norwegian universities' strategies and policies. Dashed vertical line indicates when National guidelines were introduced in 2017. Colour grading is used to indicate level of emphasis of open access with grey (low), light green (medium) and green (high). Letters signify G = General open access, E = Open access connected to education, R = Open access connected to research, 3M = Open access connected to the third mission.

Our preliminary summary is that all Norwegian universities have incorporated and strengthened their engagement in open access throughout the period. Open access is primarily seen as part of general third mission activities and often tied to science communication. Benefits of open access are not documented in annual reports.

## Cross-institutional comparison and alignment with national policy

All institutions refer directly to national documents issued by the ministry. References and quotes primarily invoke the national guidelines, but also the 2013 White Paper. Eight

institutions have revised their open access policies, inspired by the national guidelines, which is stated explicitly. For example, UiB stated in its 2020 policy that 'the government has issued guidelines for the dissemination of research data and scientific articles, which lays the foundation for the activities at UiB.'

When the national guidelines were drafted, all institutions were invited to comment. They expressed strong overall support for the guidelines. The National guidelines proposal further asked institutions to revise their policies in accordance with the guidelines, of which six complied. The remaining four wanted to introduce stronger measures than what the guidelines proposed.

As expected in a country with one research council, its influence is seen in university policies: nine institutions refer both to the Research Council of Norway and the international Plan S. NMBU and UiA explicitly write that their policies are based on Plan S, while NTNU and UiB state that the council's policy (which includes Plan S) is the foundation of their own policy.

University open access policies share many similarities, and we observe a strong coherence with the national level and the national guidelines. All current institutional policies default to green open access with opt-out possibilities for final dissemination, and all mandate depositing. Although the universities accept hybrid open access, this is only tied to financial support through national agreements and not in dedicated institutional open access funds. This is also in accordance with national guidelines.

Eight universities further state a preference for publishing in gold open access journals when researchers are faced with the choice of otherwise equal options. The last two do not state an explicit preference, but still base their policies on compliance with the national guidelines. Preferences for gold open access publishing are usually formulated cautiously and encouragingly.

As mentioned, UiO, UiB, UiT and USN connect open access to different types of local incentives or as a basis for the national results-based funding indicator. These institutions have therefore implemented mechanisms that are planned but not yet implemented at the national level.

Our observation is that national documents lend legitimacy to and anchor institutional policies, not least seen in revisions performed after the national guidelines appeared. Thus, all Norwegian Universities show a high degree of commitment to open access, and the motivation for open access as expressed in the national guidelines is unanimously shared.

## Discussion

We organise our discussion by first analysing the degree of institutionalisation and the commitment to open access in Norwegian universities. Based on our empirical findings and inspired by the institutionalisation framework in Colyvas & Powell [5] and others, we propose a framework for this process which may also serve as a theoretical starting point for later investigations. We end with a discussion of the relationship between the academic heartland [59] and open access, conclusions and suggestions for further research. In our material, we see institutionalisation playing out along three dimensions: mimesis, institutional adaptation/integration with existing missions, and maturation/commitment.

First, as documented in the previous section, we observe a *mimetic process* through which goals, perspectives and initiatives in national (and to some extent international) policy documents are incorporated into the universities' own strategic plans and other documents denoting priorities and courses of action. This is also seen over time: when national open access guidelines were introduced in 2017, university policies most often changed accordingly. There

are also many similarities in the policy details and more homogeneity than what was found in Australian institutions, where there is no national policy [56].

Mimesis is not one-way only, particularly in the first phase of national open access policy development around 2009. The higher education institutions were asked for advice from the Ministry about how to implement open access, and their suggestions became an important input into later national policy. As such, mimesis in our framework can refer to a process of mutual influence. A lot of the literature about institutional change is about external shocks, when new initiatives and developments in universities' surroundings force them to (try to) adapt. What we interpret in our case is a process of co-evolution rather than external shock, and as such not a problematic case of 'one size fits all' policy introduction as found by Kitagawa et al. [6].

University open access policies are also more subtly influenced by the practices of the Research Council of Norway. When the council introduced demands to open access publishing in contracts, universities came under pressure to strengthen their policies as contracts are signed between the council and the host institution of projects. Open access pressure through funding contracts can therefore be viewed as a more bottom-up pathway to institutionalisation. Earlier literature on institutionalisation of new initiatives in universities has often focused on top-down initiatives [e.g. 5] or on internal initiatives and the related top-down and bottom-up interplay [e.g. 5]. What the open access case may show is that national policies contain elements that instigate institutional bottom-up processes.

The 2017 national guidelines indicated that open access soon would become part of the national results-based funding indicator for the institutions. This explains why two institutions have introduced 'only open access counts' in their local reporting. Even if these (negative) incentives can be classified as a part of a mimetic process, they were introduced at a time when the national plans were not yet implemented. While there are clear traces of mimesis between the national and institutional levels, there are also differences, as shown for example by the lack of preference for gold open access at two institutions. This indicates that the universities to some extent find their own way in the implementation of open access.

Second, we see several aspects of *institutional adaptation and integration*. Incorporation of open access in the universities' main strategies is in itself a sign of institutionalisation, which can be seen a proxy for institutional support of change [3]. Open access has increasingly become a natural part of how the universities present their societal relationship and is thus perceived (or at least accepted) as a direct means of knowledge transfer and/or as part of an obligation to share knowledge with society. However, the general formulations in the documents indicate that the benefits of free access to the scientific literature are somewhat unclear with respect to external user groups.

Universities connect open access less directly to innovation and commercialisation compared to general societal linkages. This could be because innovation-oriented university-industry relationships are institutionalised through research partnerships or other processes. The 'entrepreneurial university' does not yet give a central role to the scientific literature, but puts more emphasis on personal, informal and tacit exchange of knowledge [4, 80]. This means that the relationship between innovation and open access is still weakly developed, even if national and international polices argue that innovation benefits from open access. This could be viewed as examples of copying arguments given in national polices, and thus a part of isomorphism.

The absence of arguments for open access as supportive of education is equally interesting, especially since the open access movement's initial vision was that it would strengthen both research and teaching. While the Budapest declaration emphasised education and downplayed importance for the third mission, the institutional strategies and policies do the opposite. One

explanation may be that the institutions view employees and students as a single user group; benefits for students are taken for granted if access is achieved for researchers. Students are considered a part of the open access policies through obligations to deposit their work, and their otherwise absence from open access policies can also be explained by the strong role of university libraries in open access practices. Librarians are often in charge of the institutional repository and support like online information, rights clearance and open access funds. The libraries' intimate ties to student activities and open access may seem too obvious to be mentioned in institutional strategies.

It is also interesting that strategy documents emphasise open access links to a broad version of the third mission rather than research. The explanation could be that strategy documents are important for signalling priorities to stakeholders outside academia, while open access policies are internal communications to employees. The many connections claimed between open access and the third mission also suggest little resistance and conflicts concerning the introduction of open access, as opposed to the resistance and conflicts seen in many places when the third mission was promoted heavily [7]. Open access tensions are of a different nature than opposition to the policy, which will be discussed below.

The lack of any examples of concrete benefits for non-academic users contributes to the generic nature of claims about positive effects. There is a notable gap between 'theory and practice', adding to the lack of evidence for the direct usefulness of open access for the third mission [41]. This may also explain why local policies default to generic arguments for open access where science is portrayed as a public good [47]. A pessimistic interpretation is that the benefits are exaggerated and merely echo broad policy aims, or that the core of the open access movement is primarily a transformation of the academic publishing sector, where societal benefits serve as an opportune legitimation. On a more optimistic note, it may be claimed that such benefits are often indirect and long-term and therefore difficult to document and attribute [41].

Relatedly, the low level of systematic and detailed reporting of open access in annual reports may be seen as puzzling. Unlike the challenging task of documenting benefits for non-academic users, progress of local open access levels and shares would be easier to document even if some aspects of open access could be difficult to estimate. Reporting cycles provide an explanation as annual reports mostly are published in the first quarter of a year, while the official reporting of publications in Norway takes place in April. We also assume that lack of systematic and detailed open access reporting is connected to the absence of consistent demands of these numbers from the Ministry, which publishes separate reports about open access.

Finally, there is a *maturation and commitment*. We see the removal of opt-out mechanisms and clearer demands to faculty in institutional policies as a sign of progression and recognition of what constitutes an effective policy [53]. The institutions convey more strongly voiced expectations to, and responsibility for, open access, also signalling its importance through mandates.

Different traditions or ways of making decisions in the ten Norwegian universities could be the reason why two universities have framed open access primarily as an intellectual property rights issue and therefore relevant for formal treatment in employment contracts. Others have focused on carrot/stick aspects like prioritising open access publication in promotion decisions, which also is connected to effective policies [53]. We find these differences noteworthy although they may all be roads to the Rome of open access. The University of Oslo later introduced negative incentives, which suggests that enforcing open access through working contracts had limited effect.

The introduction of local incentives at four institutions are proactive measures that arguably show great commitment. Including open access in evaluations for promotion and other

**Table 3. Framework for two stages–early and late for the 2009–2021 period–of open access institutionalisation across three dimensions: Mimesis, institutional adaptation and integration, and maturation and commitment.**

| Main dimension | Sub-dimensions | Stages of institutionalisation | |
|---|---|---|---|
| | | Early stage | Late stage |
| Mimesis | *Anchoring/Compliance* | Few or weak (and prosaic) references to national policy including white papers. | Alignment with/ extensive compliance with national policy. References to international policy. |
| Adaptation and integration | *Adaption* | No presence of open access in main strategy/ sub strategies. | Accepted place in strategies. |
| | | | Distinct part of the university's main goals. |
| | *Integration* | Tied only to research. | Tied to all existing main tasks/societal role. |
| | | | Improved ties to research mission by moving towards open science. |
| | *Documentation (Legitimacy/Reporting)* | No demonstration of positive effects. | Stated goals of progress. |
| | | No reporting. | Open access indicators/detailed reporting. |
| Maturation and Commitment | *Mandates* | Voluntary behaviour and laissez-faire. | Mandatory with few or no opt out clauses including use of 'must' and similar language. |
| | | Ask 'politely' in policies. | Shorter embargo preferred/preference for immediacy. |
| | | Timeliness unspecified. | |
| | *Responsibility* | Responsibility for open access unclear. | Shared responsibility between institution and researchers. Researchers are expected to know the rules. |
| | *Incentives* | No local incentives/negative incentives. | Tied to tenure, promotions, sabbaticals, distribution of internal research resources. |
| | *Boundaries* | Open access and academic freedom/legal rights and licensing not discussed. | Clear positioning/borders with academic freedom/legal rights for individual researchers and respect for the academic heartland. |
| | | | Author retention rights strategies. |

decisions signals a growing acceptance of the important role research evaluation has for open access practises [81]. However, it is not clear whether the incentives are put into effect. There may also be institutional cultural differences where incentives of this kind are less controversial in some institutions or that introducing such incentives is an attempt to create a local change.

An important category in our data concerns how institutions deal with open access and academic freedom. Six institutions discuss boundaries or tensions between these two, and all grade academic freedom higher. For the last four universities, academic freedom was graded higher than commercialisation, affirming the important status of academic freedom.

We end this section with a proposal for a framework for understanding the facets of institutionalisation at the institutional level, presented in Table 3. Following the suggestions of Colyvas & Powell [5] we distinguish roughly between an early and late state of institutionalisation–as the process is still ongoing with no definite signs that open access is yet taken for granted (although the universities are certainly moving in that direction). Our examples of the institutionalisation process in the cells also represent important general summaries of our empirical material. These characteristics partly reflect the ones in Colyvas & Powell's study of commercialisation at Stanford University but also contain unique aspects of the open access case.

In our framework, the institutionalisation process has three main dimensions as discussed above: mimesis, institutional adaptation and integration, and maturation and commitment.

## The academic heartland and open access

We have found coherence among university policies and strong links to national policies. This is not merely a process of adaptation and compliance–universities have been formally asked for advice and have been able to influence the national policies. As such, the idea of open access has probably also become institutionalised in the policy system, including the Ministry and research funders.

Still, we observe tensions surrounding open access, where Plan S is accused of interference with academic freedom [13]. Open access has wide support, but not when perceived as in conflict with core values in academia, captured in the expression 'the academic heartland' [59]. Clark noted tensions and conflicts with the introduction of third mission activities (ibid.), where stimulating collegiality, autonomy, academic freedom and achievement was of critical importance for institutionalisation. We argued in the introduction that such tensions are less likely in the open access case, since open access is more in alignment with scientific norms and traditional missions. Our main empirical findings offer support: national open access guidelines were largely applauded by the universities.

However, this is not the case with Plan S. The most relevant difference is that the Norwegian national open access guidelines do not view any (type of) journal as ineligible, while Plan S excludes the choice of journals that do not meet the requirements of the policy. In addition, the legal aspects of the exclusion of journals is considered within the law in the case of Plan S, as it ultimately is a voluntary arrangement to sign Plans S contract agreements [70]. This is not the case when scientists perform research as regular employees according to Norwegian legislation, and this is the domain of institutional policies. In the foundational texts of current legislations, it is stated explicitly that the choice of journals belongs to the researcher (ibid.).

In a broader perspective, this consideration helps explain the profile of local and national policies. Open access is not simply an issue of academic ideals and Mertonian norms, but also of academic freedom and norms protected by (Norwegian) law, and many universities have pushed for an even stronger formal and legislation-based protection of academic freedom. There are thus limits to how strict and compulsory rules of open access can be designed. No institution ultimately mandates open access. There are opt-out-options available even though the researcher must comply for the benefits of open access to become realised.

This reasoning is also found in our material on the discussions around academic freedom and open access and echoes that researchers' choice of publication channel is regarded a part of their individual choice as university staff. The centrality and legal status of academic freedom naturally leads to opt-out clauses and explains why gold open access publishing at best is stated as a preference.

Legal matters are also the reason policies contain opt-out clauses for final dissemination in the case of green open access. This is to avoid situations where researchers run out of options if co-authors or publishers refuse to grant permissions for green open access. Instead, the first initial step of depositing is made mandatory as depositing poses conflict neither with legal considerations nor with academic freedom. These considerations are not only found at the institutional level, they are also part of the national guidelines [68].

We interpret this as a confirmation of the co-evolutionary aspect of the institutionalisation process. National guidelines and governmental decrees have respected the 'academic heartland' in all policy iterations and sought advice from the scientific community when implementing them. We therefore find fewer tensions in the introduction of open access as expressed in the national guidelines, compared to the introduction of the third mission [6–8, 14, 15]. The influence of Plan S on some of the polices was unexpected mainly because of the controversies surrounding the policy. However, it seems Plan S has served as an inspiration for institutional policies rather than as a script for detailed implementation. The academic home remains protected, and we interpret the influence of Plan S as a sign of commitment.

## Conclusion

In this paper, we have used a qualitative and inductive analysis of documents to study how Norwegian universities responded to expectations about open access and implemented them

in internal institutional policies. The documents are issued in a period of increasingly strong political interest in open access both nationally and internationally. During this period, we see movements in how Norwegian universities handle open access and find that:

1. Open access is becoming increasingly institutionalised in Norwegian universities. It is incorporated in strategies, albeit in a very general manner, and primarily seen as supportive of dissemination and societal contact. However, the universities do not demonstrate this with evidence.

2. The universities show a high degree of commitment with revisions of local polices following national policy, sometimes with stronger measures than what the Ministry issued.

3. Introduction of open access happens without challenging the academic heartland, particularly with respect to the choice of publishing outlet. Academic freedom is the cornerstone of the research mission and outweighs other considerations, as has been found in previous studies [5, 6]. One important backdrop is that academic freedom is protected by law in Norway.

4. Rather than a contested pattern of institutionalisation following problematic national demands, as described in many of the cases of the third mission [e.g. 5–8], our case represents a more co-evolutionary process where universities have exercised influence on national guidelines.

To conclude, the introduction of open access in Norway can be characterised as rather smooth and successful, with little friction between the institutional and national level and a development over relatively long time. A close relationship between the universities–which are public–and the Ministry, and the avoidance of top-down-instructions only, has led to institutional engagement. This is also mirrored in practises and the overall levels of open access (see S1 Fig). Norway is a small country with relatively few universities and a few central organisations that govern research. Although universities compete for attention and external funding, their funding streams are similar and from the same sources. We would therefore not necessarily expect identical results in countries were the conditions are different and/or heterogeneous institutions are engaged in fiercer competition on economic grounds. The study is also deliberately limited to universities and have excluded other types of research performing institutions. Still, the rest of Norway's university colleges (some of which are private) and the research institutes (public and private) also largely depend upon funding from the same sources, at least for their research activities.

We nevertheless claim that the study can be useful for analysing developments in other countries, and our proposed theoretical framework opens up also for understanding differences and heterogeneity. The dimensions and subdimension in the framework hold a potential for understanding and interpreting movements between stages of institutionalisation, even though we assume that local contextual characteristics may require some adjustments to the framework.

A main limitation of our study, perhaps also a strength, is that we take policies and strategies as a de facto expression of the institutions' views and values. However, formulations in policies and strategies are results of creative processes, human factors and negotiations between many contributors–these are both 'hidden' and disregarded. We nevertheless claim that documents provide high validity for our aims.

There are several topics that calls for more investigation. First, institutional policies and practises are still changing, which means further monitoring may provide valuable. We have not explored mimesis between the institutions but have found references in the documents

suggesting that this is happening. There are also both formal and informal collaborations between universities at different levels, including matters concerning open access, for example through library collaboration. We believe these factors also contribute to the shaping of policies and behaviour at universities. Identifying these mechanisms would provide valuable and deeper insights in the processes of institutionalisation and how they engage various professional networks. Further, while we have concentrated our investigation on open access, there seem to be a movement towards more broad open science policies that include open access as one of several elements. How broader policies for open science plays out beyond open access is an interesting topic for later work.

Finally, the role of incentives and research evaluation in policies is a fruitful research topic. This include how incentives and evaluations are introduced, shaped, and how they affect publishing behaviour. A key to open access is connected to research evaluation, which poses a direct change in the structures of the academic heartland. This may prove more challenging than introducing the principles of open access.

## Supporting information

**S1 Fig. Norwegian universities uptake of open access in the period 2013–2020.** Overview of gold open access, hybrid open access and green open access in the period 2013–2020 given in percentage. Upper line indicate growth in total levels of open access, bottom line indicate growth in gold open access.
(TIF)

## Author Contributions

**Conceptualization:** Lars Wenaas, Magnus Gulbrandsen.

**Data curation:** Lars Wenaas, Magnus Gulbrandsen.

**Methodology:** Lars Wenaas, Magnus Gulbrandsen.

**Writing – original draft:** Lars Wenaas, Magnus Gulbrandsen.

**Writing – review & editing:** Lars Wenaas, Magnus Gulbrandsen.

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
