## [Decision Letter · Decision Letter 0]

18 Apr 2022

PONE-D-22-07577The green, gold grass of home: introducing open access in universities in NorwayPLOS ONE

Dear Dr. Wenaas,

Thank you for submitting your manuscript to PLOS ONE. After careful consideration, we feel that it has merit but does not fully meet PLOS ONE’s publication criteria as it currently stands. Therefore, we invite you to submit a revised version of the manuscript that addresses the points raised during the review process.

ACADEMIC EDITOR: This is a neat research contribution to the field with results about Norway as a case country with regard to the rising demand of open access. The reviewers also found it strong and convincing in general, and raised several issues you are requested to look at and deal with thoroughly. In particular, as Reviewer 1 underscores, the structure of the paper should be re-thought at certain points so that it possesses more consistency. As Reviewer 2 points out, you should be more concrete about why you worked with these ten universities of your choice and place them in the larger context of Norwegian higher education. Both reviewers commented on the literature review, probably a separate section on this would help. Please, in your rebuttal deal with all of their comments.

We look forward to receiving your revised manuscript.

Kind regards,

István Tarrósy, PhD

Academic Editor

PLOS ONE

Journal Requirements:

[I have read the journal's policy and the authors of this manuscript have the following competing interests: Lars Wenaas is a PhD-candidate at the TIK-center at the University of Oslo while holding a position at 'Sikt', a governmental body reporting to the ministry of education and research in Norway. The position is a part of the department with responsibility of coordinating open access affairs in Norway.]

4. We note you have included a table to which you do not refer in the text of your manuscript. Please ensure that you refer to Table 1 in your text; if accepted, production will need this reference to link the reader to the Table.

Reviewers' comments:

Reviewer's Responses to Questions

**Comments to the Author**

1. Is the manuscript technically sound, and do the data support the conclusions?

Reviewer #1: Yes

Reviewer #2: Partly

2. Has the statistical analysis been performed appropriately and rigorously? 

Reviewer #1: N/A

Reviewer #2: N/A

3. Have the authors made all data underlying the findings in their manuscript fully available?

Reviewer #1: Yes

Reviewer #2: Yes

4. Is the manuscript presented in an intelligible fashion and written in standard English?

Reviewer #1: Yes

Reviewer #2: Yes

5. Review Comments to the Author

Reviewer #1: General remarks

The manuscript aims to give a review of the institutionalization processes of Norwegian universities as those react to open access publishing. The authors conducted exploratory research to uncover how the universities implement changes and apply new practices to give institutional responses to open access policy.

The methodology is chosen and applied correctly. The document analysis is conducted consistently and described clearly in the manuscript. Both the advantages and concerns regarding the institutional processes of open access are highlighted.

This manuscript is actual, interesting, and reflects valuable research about open access.

However, some issues of open access that are relevant from the universities point of view are not discussed in the manuscript. Thus, the reviewer has some remarks and suggestions to improve the manuscript.

Major issues

1. The topics of the three sections of the literature review are separated strongly. So, there is a considerable thematic gap between the consecutive parts. The content of the sections should be linked consistently. It is also worth considering whether the sequence of the three sections of the literature review is appropriate.

2. In the second section of the literature review, the authors should also address how open access and the quality (or rankings) of academic journals are related to each other and whether there are any differences among disciplines regarding open access publishing.

3. The authors found little information about the advantages of open access for education in the studied documents. However, more details should be provided about how open access can support education. Some statements and remarks would be useful and may be incentive.

4. Why open access publishing can be advantageous for researchers should also be pointed out.

5. The manuscript does not explain the issues of open access publishing funds. However, these are also relevant for universities and researchers (as the authors of papers). How universities cover the article processing charges (APC) and distribute the (dedicated) open access publishing funds among disciplines, departments, research projects, and researchers are crucial questions. It is also relevant how researchers can require financial resources to cover APC. These issues are also related to institutionalization processes and should be discussed in the manuscript.

Minor issues

1. The main title of the manuscript should be worded more professionally.

2. The time frame of the study should be indicated in the abstract.

3. Single and double quotations marks are applied alternately in the text. A coherent format is required.

4. The page number is missing in some word by word citations, e.g. on page 5, in lines 191-193. It is required in each quotation.

5. The authors should give some remarks in the conclusion that could be useful for other countries and universities.

6. Notes should be provided in Figure 1 to understand the markings clearly.

7. In Figure 2, all the signs have to be given in English.

All in all, this is a valuable manuscript. It could be improved significantly by considering the abovementioned remarks and suggestions.

Reviewer #2: The paper is focusing a very up-to-date and timely topic as open access phenomenon is with us just since few years. Thy analysis provides a fair overview of the topic, especially the context (its connection with the universities' third mission). Investigating strategic documents of different universities is a great idea, however, it should be emphasized better, why not all Norwegian universities are included. Also, the literature review part should be established. Now we can see the theoretical framework but not an overview, what other authors wrote about this topic. It would be an essential issue, as the topic itself is very young. So, in this way, a critical-analytical, comprehensive literature review would be one of the strengths of the paper and could generate a more wide publicity.

6. PLOS authors have the option to publish the peer review history of their article (what does this mean?). If published, this will include your full peer review and any attached files.

Reviewer #1: No

Reviewer #2: No

---

## [Author Response · Author response to Decision Letter 0]

8 Jul 2022

Please see 'Response to Reviewers.docx' for a complete response to the comments from editor and reviewers.

---

## [Decision Letter · Decision Letter 1]

3 Aug 2022

The green, gold grass of home: introducing open access in universities in Norway

PONE-D-22-07577R1

Dear Dr. Wenaas,

We’re pleased to inform you that your manuscript has been judged scientifically suitable for publication and will be formally accepted for publication once it meets all outstanding technical requirements.

Kind regards,

István Tarrósy, PhD

Academic Editor

PLOS ONE

Additional Editor Comments (optional):

The author has revised the original manuscript following the critical remarks and suggestions of the reviewers, which then resulted in an even more coherent and convincing paper. I find it relevant with a novum aspect sufficient for publication without requesting further changes or modifications.

Reviewers' comments:

Reviewer's Responses to Questions

**Comments to the Author**

1. If the authors have adequately addressed your comments raised in a previous round of review and you feel that this manuscript is now acceptable for publication, you may indicate that here to bypass the “Comments to the Author” section, enter your conflict of interest statement in the “Confidential to Editor” section, and submit your "Accept" recommendation.

Reviewer #1: All comments have been addressed

Reviewer #2: All comments have been addressed

2. Is the manuscript technically sound, and do the data support the conclusions?

Reviewer #1: Yes

Reviewer #2: Yes

3. Has the statistical analysis been performed appropriately and rigorously? 

Reviewer #1: N/A

Reviewer #2: Yes

4. Have the authors made all data underlying the findings in their manuscript fully available?

Reviewer #1: Yes

Reviewer #2: Yes

5. Is the manuscript presented in an intelligible fashion and written in standard English?

Reviewer #1: Yes

Reviewer #2: Yes

6. Review Comments to the Author

Reviewer #1: General remarks

The authors revised their manuscript considerably. The literature review has been improved consistently. The manuscript has also been carefully corrected based on minor issues.

All in all, the revised manuscript is relevant and valuable.

Reviewer #2: The authors improved the paper about my previous recommendations. The paper represents a higher academic quality than the previous version. I can accept this version of the paper for publication without further changes.

7. PLOS authors have the option to publish the peer review history of their article (what does this mean?). If published, this will include your full peer review and any attached files.

Reviewer #1: No

Reviewer #2: No

---

## [Editor Report · Acceptance letter]

8 Aug 2022

PONE-D-22-07577R1 

The green, gold grass of home: introducing open access in universities in Norway 

Dear Dr. Wenaas:

I'm pleased to inform you that your manuscript has been deemed suitable for publication in PLOS ONE. Congratulations! Your manuscript is now with our production department. 

Kind regards, 

on behalf of

Dr. István Tarrósy 

Academic Editor

PLOS ONE